# Generative adversarial network with Hierarchical semantic Prompt Constrainting CLIP for high-quality Text-To-Image Synthesis

## Abstract

How to synthesize efficient, controllable, and semantically relevant high-quality images based on text is currently a very challenging task. Combining generative adversarial networks with CLIP models to improve the quality of synthesized images has revitalized GAN in the field of generation. Compared with diffusion models, GAN has faster generation speed, fewer training resources and parameters, and more controllable generation results. However, the current methods for combining CLIP and GAN are relatively rough, mostly used as text encoders and feature bridges, without fully utilizing the semantic alignment ability of CLIP networks, ignoring the structural and hierarchical nature of semantic features, and resulting in poor semantic consistency in synthesized images. In response to these problems, we propose HSPC-GAN, which is a method of constructing structural semantic prompts and using them to hierarchically guide CLIP to adjust visual features for generation of high-quality images with controllable semantic consistency. HSPC-GAN extracts semantic concepts through part of speech analysis, constructs a prompt generator and a prompt adaptor to generate learnable hierarchical semantic prompts, and using these prompts to selectively guide CLIP adaptors to adjust image features to improve semantic consistency between synthesized images and conditional texts. At the same time, we introduced the mining of hard negative samples into the construction of the discriminator loss function for the first time, improving the discriminator's ability to distinguish mismatched samples and reducing the impact of the generated model's requirements for batch size and epoch on training results. A large number of experimental results have proven the effectiveness of our method, which can quickly synthesize high-quality images with consistent semantics, and achieve SOTA score on public datasets.

## 1 Introduction

Recently, the field of generative models has undergone tremendous changes with the introduction of large-scale pre-trainde autoregressive models (Rombach et al., 2021; Nichol et al.,2021; Yu et al., 2022; Saharia et al., 2022) and Diffusion Model (Ramesh et al., 2021; Robin et al., 2022). These methods demonstrate the synthesis performance far beyond traditional methods (Tao et al., 2022; Liao et al., 2022) and many appealing applications based on them are appearring. Research on related works has also received more attention, especially for text-to-images synthesis.

How to synthesize efficient, controllable, and semantically relevant high-quality images based on context is currently a very challenging task in the field of text image generation. Although large pre-trained multimodal models and diffusion models can synthesize images that look very realistic, they rely too heavily on large-scale training data, massive parameters, and artificial prompts and adjustments for the generation process. And the slow reasoning speed and huge computational resource overhead are also very daunting. Similarly, generative adversarial network (GAN) has faster generation speed and more smooth latent controllable training space, which enables more controllable synthesis, but the quality of generated images is not not excellent. To generate high-quality text-images more quickly, some recent work (Kang et al., 2023; Tao et al., 2023) began to

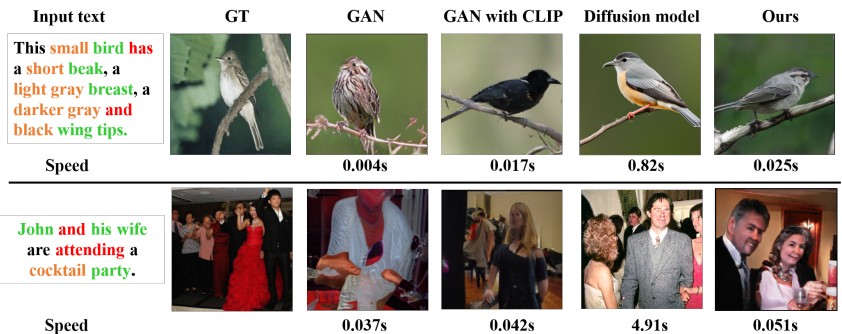

Figure 1: Generation examples in CUB and COCO. Our model can generate images with higher-semantic-consistency, compared with groud truth and other generation methods include the traditional GAN (SSA-GAN), GAN with CLIP(GALIP), Diffusion Model(LDM). Speed is calculated on 1x3090 GPU

combine GANs with CLIP models (Radford et al., 2021; Li et al., 2022; Yu et al., 2022; Wang et al., 2022) which have a capability of aligning cross-modal semantic. However, these methods are relatively rough for the use of CLIP, which is mostly used as text encoders or feature bridges, without fully utilizing the semantic alignment ability of CLIP. These methods neglect to consider the semantic structure and hierarchy of the existence in the text description, and ignore the specificity of named entities, which results in poor semantic consistency of the synthesized image with text.

In response to slove the above problems, we propose a novel method of using CLIP in GAN for text-to-image generation, named HSPC-GAN. CLIP is no longer just a tool for cross modal feature transformation, but rather participates in visual feature generation and adjustment hierarchically based on prompt guidance. Instead of directly encoding and transforming conditional contextual information into visual features for generation, we extract structured semantic concepts from text descriptions, and construct learnable local prompts containing entity, attribute, and relationship informations. These prompts guide the CLIP adaptor to adjust image features hierarchically together with global prompts containing complete context, which can enable GAN to synthesize semantically related images in a more refined manner.

In this work, we introduce how to construct structured semantic prompts through a prompt generator Fig.3 and prompt adaptor Fig.4(a) . We use natural language processing tools (Qi et al., 2020) to analyse conditional text, include phrase segmentation, part-of-speech and named entity recognition. For "John and his wife are attending a cocktail party",is marked as entyties "person john, wife, party ", attributes "cocktail ", relationships "and, attend ". After padding semantic patches, textual concepts are fused by controling through text semantic scene maps, then construct local prompts by computing attention with global text. We use local prompts and global prompts to guide the frozen-parameter CLIP's transformer block hierarchically adjust feature maps. Global prompts are added at both ends of the prompt sequence to ensure that global semantic information is not lost. At the same time, each prompt independently selects whether to replace the class token, to avoid redundant prompts problem for text with different amounts of information.

In addition, referring to the resemblance of cross-modal retrieval task(Faghri et al., 2017), we novely design the Hard Mining Matching-Aware Loss (HMMAL) for mismatch sample in one batch, which can accelerate our training speed and improve the semantic consistency of generated image and conditional text. This loss can be generalized to apply most generation methods involving semantic matching. We will do more work in the future to prove this viewpoint.

Furthermore, HSPC-GAN has faster synthesis speed, fewer computing resources and higher semantic consistency between images and texts compared to diffusion and autoregressive models. As shown in Fig.1, compared to other GAN with CLIP methods, our method is able to focus on more detailed information in text and have a stronger ability to synthesize complex real images. Numerous experiments on publicly datasets have proven the effectiveness of our method. In COCO dataset,

our FID score has decreased to 4.97, and our CLIP Score has reached 0.3507. The CLIP Score of testing in CUB dataset has reached 0.3306.

- We propose an intuitive and effective text-to-image synthesis method which guides CLIP adjustment of visual features hierarchically, by using Structural semantics prompts. The method can synthesize highly semantically relevant text images.

- We propose a method for constructing structured semantic prompts from the conditional text. This method can fully extract fine-grained semantic information for guiding CLIP.

- We introduce difficult negative mining (Hard Mining) into the discriminative loss of the generated model for the first time, which can accelerate model training and improve semantic consistency of generated images and texts.

- Extensive experiments demonstrate the effectiveness of our proposed HSPC-GAN, which achieves the state of art on multiple public datasets.

## 2 RELATED WORK

**GAN-based image synthesis** GAN-based image synthesis often necessitates the determination of the underlying probability distribution corresponding to given images, a process commonly referred to as inversion (Zhu et al., 2016: Xis et al, 2021). One prevailing approach in the GAN-based Image Synthesis is based on optimization techniques (Abdal et al., 2019; 2020; Zhu et al., 2020; Gu et al., 2020). This involves iteratively adjusting latent variables to synthesize images that match the given input image, effectively reversing the generative process. Optimization-based inversion methods offer flexibility, particularly when handling novel or unseen concepts. Alternatively, an encoder-based approach has been employed (Richardson et al., 2020; Zhu et al., 2020a; Pidhorskyi et al., 2020; Tov et al., 2021) to perform image-to-latent space mapping. Encoders aim to learn a mapping function that can project images into the latent space of the GAN, effectively enabling image manipulation within the learned embedding space. However, encoders face more stringent generalization requirements and often necessitate training on extensive, web-scale datasets to achieve comparable versatility as optimization-based methods. In our work, we provide a detailed analysis of our embedding space in the context of the GAN inversion literature, elucidating the fundamental principles that persist and those that have evolved over time. This analysis contributes to a deeper understanding of the strengths and limitations of different inversion techniques within GAN-based image synthesis.

**CLIP Encoder** Compared to unimodal vision feature extraction methods like CNN (Liu et al., 2022; Shen et al., 2023), Vision Transformer (Dosovitskiy et al., 2020; Liu et al., 2021) and hybird (Dai et al., 2021; Tu et al., 2022), CLIP stands out as a multimodal feature extraction approach (Radford et al., 2021; Li et al., 2022; Yu et al., 2022; Wang et al., 2022) that achieves enhanced generalization capabilities. CLIP's pretraining on large-scale image-text datasets, coupled with contrastive learning, facilitates the creation of a joint latent space, particularly advantageous in zero-shot and few-shot downstream tasks.

**Prompt in generation** In the field of NLP, the use of prompts has gained significant popularity. With prompt design(Liu et al., 2021), Large-scale pretrained models have demonstrated impressive generalization capabilities without the need for fine-tuning. In recent years, Large Language-Image (LLI) models(Saharia et al., 2022; Ramesh et al., 2022; Rombach et al., 2022) have exhibited remarkable semantic generation and compositional abilities, garnering unprecedented attention from both the research community and the public. However, achieving optimal generation results with these large-scale LLI models often requires the use of complex prompts. Compared to the manually designed (Radford et al., 2021) and discrete prompts (Jiang et al., 2020; Gao et al., 2021c) , employing learned continuous prompts (Zhou et al., 2022; Gao et al., 2023; Khattak et al., 2023) tailored for specific tasks has proven to be more effective in eliciting superior performance.

## 3 METHOD

In this section, we start with the contruction of learnable hierachical semantic prompts, introduce how to generate hierachical semantic prompts from conditional context, and adjust discrete prompts into continuous learnable vectors. Next, we descibe the concrete technical details of how we use

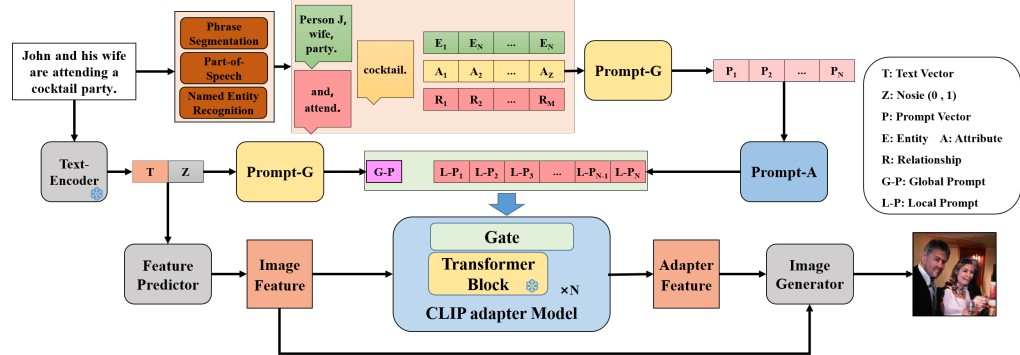

Figure 2: The architecture of the Generator. It includes text concept mining module, prompt generator and adaptor , feature predictor, CLIP feature adaptor, text encoder, and image generator.

hierarchical semantic prompts to guide the frozen parameter CLIP-VIT to update the coarse-grained visual features, and generate fine-grained visual features with hierarchical semantic information. Finally, we discuss the impact of difficult negative sample mining on the discriminative loss of generate adversarial networks..

### 3.1 GENERATOR

The architecture of our HSPC-GAN's generator is shown in Fig.2. We follow the end-to-end basic structure proposed in (Ming et al., 2023), but add text concept mining, the hierarchical semantic prompt generator and adaptor, replace their CLIP-VIT module with our CLIP-Adaptor module, and adjust the image generation blocks.

#### 3.1.1 TEXT CONCEPT MINING

We annotate the conditional context with text concept mining, using natural language processing tools (Qi et al., 2020) which can perform tokenization, phrase segmentation, named entity recognition, part-of-speech & morphological features tagging et al. on a piece of text description. Through NLP tools, we convert complex descriptive text into a form that is easy to understand by text encoder, such as "George Johnson" being labeled as "Person" and "Stanford" being labeled as "organization". So, the complex text description is transformed into collections of entities, attributes, and relationships to represent all semantic conceptual information. They are represented entity sets $\mathbb{E}\{e_1, w_2, ..., e_n\}$, attributes sets $\mathbb{A}\{a_1, a_2, ..., a_z\}$, and relationship sets $\mathbb{R}\{r_1, r_2, ..., r_m\}$. Meanwhile, this conditional context information is fed into a text encoder for encoding to obtain text vector $t$. We adopt the pre-trained text encoder provided by (Radford et al 2021). In many existing works, this text encoder is generally adptopted by fixing its parameters, because relevant experiments (Xu et al. 2018) have proven that the fine-tuning text encoder does not improve the performance of text-to-image synthesis.

#### 3.1.2 HIERARCHICAL SEMANTIC PROMPT GENERATOR

Prompt has a significant improvement effect on controlling and adjusting the generation target in the generation task (Li et al., 2021; Li et al., 2022; Robin et al., 2022 ). Inspired by Prompt's application in text generation tasks (Li et al., 2022) and text classification tasks (Ronald et al., 2021), we design a hierarchical semantic prompt generator to replace the method of directly predicting prompts based on sentence and noise vectors through an FC layer. The generator module is shown Fig.3. It consists of three parts: a text encoder, a scene graph parser, and a hierarchical semantic fusion gating controller.

The entity set $\mathbb{E}$, attribute set $\mathbb{A}$, and relationship set $\mathbb{R}$ are respectively input into the text encoder for encoding to vectors $\boldsymbol{V}_E, \boldsymbol{V}_A, \boldsymbol{V}_R$. Before being encoded, entities are added with a semantic prompt patch "a photo of a " to strengthen entity attributes. The others are also similar. The reason for this is that the training process of CLIP adopts the form of semantic prompt patch added label for training,

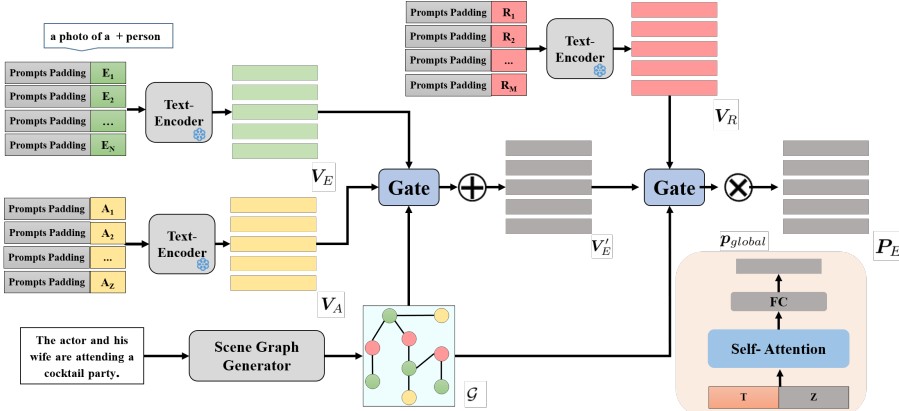

Figure 3: The architecture of the Hierarchical Semantic Prompt Generator.

which can better align the text features described by each entity through semantic patches. This has also been proven in our subsequent experiments.

The conditional context is fed into the semantic scene graph parser (Peter et al., 2016) to generate a text scene graph $\mathcal{G}$, which is used to control the fusion of hierarchical semantics. When entity vector $\boldsymbol{v}_{e_i}$ and attribute vector $\boldsymbol{v}_{a_j}$ pass through the gate of control, they are queried in the semantic scene graph whether there is a connection between the entity $e_i$ and the attribute $a_k$. If there is an edge between this entity $e_i$ and this attribute $a_j$, they are weighted together to generate additional attribute entity vector set $\boldsymbol{V}'_E$. Similarly, for the relational features vector set $\boldsymbol{V}_R$ and the attribute entity vector $\boldsymbol{v}'_{e_i}$, when there is a tuple $(e_i, r_{ij}, e_j)$ present in the scene graph $\mathcal{G}$, the inner product of the relationship vector $\boldsymbol{v}_{r_{ij}}$ and entity vector $\boldsymbol{v}'_{e_j}$ will be used to update entity feature vector $\boldsymbol{v}'_{e_i}$. In this way, we define the entity prompts $\boldsymbol{P}_E$ generation:

$$\boldsymbol{V}'_E = f_{gate}(\boldsymbol{V}_E, \boldsymbol{V}_A, \oplus)$$

$$\boldsymbol{P}_E = f_{gate}(\boldsymbol{V}'_E, \boldsymbol{V}_R, \otimes)$$

$$\boldsymbol{p}_{e_i} = \boldsymbol{v}'_{e_i} + \sum_{j,(e_i,r_{ij},e_j)\in\mathcal{G}} \boldsymbol{v}_{r_{ij}} * \boldsymbol{v}'_{e_j}, \tag{1}$$

$f_{gate}$ represents a fusion gate controller according to the sence scene graph. For the global prompt, we concatenate the conditional context vectors $\boldsymbol{t}$ with a random noise $\boldsymbol{n}$ to generate global contextual semantic prompt $\boldsymbol{p}_{global}$ through a self-attention layer and a FC layer.

### 3.1.3 LOCAL PROMPT ADAPTOR

The module of the prompt adaptor is to transform the prompts $\boldsymbol{p}_e$ into learnable local prompt vector features $\boldsymbol{p}_l$ (Fig 4a). We add the position embedding vector $\boldsymbol{v}_{pos}$ of the entity in the sentence to the entity prompt vector $\boldsymbol{p}_e$, and the generation of position embedding vector using the method (wang et al. 2020). To a entity prompt $\boldsymbol{p}_{e_i}$:

$$\boldsymbol{p}'_{e_i} = \boldsymbol{p}_{e_i} + g_i(pos) \tag{2}$$

$g_i$ is a function of embedding the entity $e_i$ position from its word index. We calculate the cross-attention between prompt vector $\boldsymbol{p}'$ and context vector $\boldsymbol{t}$ to update $\boldsymbol{p}'$, and also through an FC layer to learn local prompt $\boldsymbol{p}_{local}$.

### 3.1.4 HIERARCHICAL SEMANTIC PROMPT CLIP ADPTOR

The predicted hierarchical semantic prompts are used to guide CLIP to adapt the fine-grained visual features. The pretrained CLIP-ViT has the ability to obtain visual concepts from text features and can bridge the gap between modal semantics. We design a brand new structure which can hierarchically extract different visual concepts from the CLIP of frozen parameters to adjust the image feature

Fig.4b. We input semantic prompts in the order of global, local, and global, and use gating to adaptively determine whether to replace class tokens to obtain visual information from CLIP. The reason for this is that the number of target entities in the context is limited and different, and not all local prompts can work. Global prompts are used to ensure that all information in the text is taken into account and local prompts guide personalized generation.

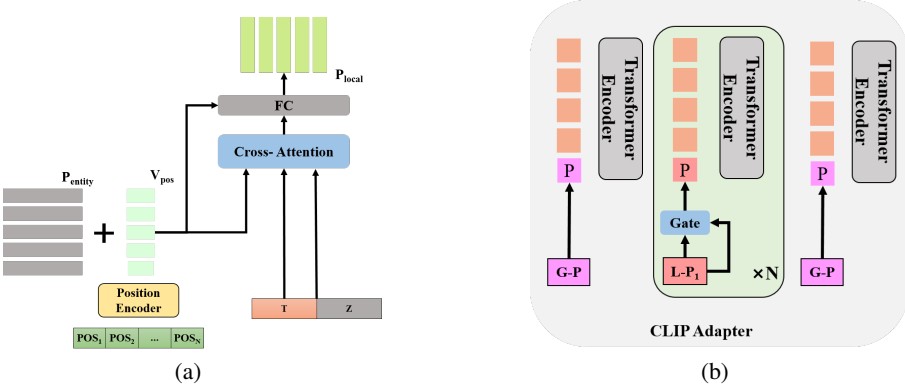

Figure 4: (a)The architecture of the Prompt Adaptor. (b) The architecture of the Hierarchical semantic CLIP Adaptor

## 3.2 DISCRIMINATOR

### 3.2.1 BASIC OBJECTIVE FUNCTIONS

The adversarial loss is used to train the ability of the generator and discriminator. For the basic loss, we follow the function designed by (Tao et al., 2023). It adopts the hinge loss (Zhang et al., 2019) and one-way discriminator (Tao et al., 2022). The basic formulation is shown as follows:

$$
\begin{aligned}
L_D = &-\mathbb{E}_{x\sim\mathbb{P}_r}[min(0, -1 + D(x, e))] \\
&- (1/2)\mathbb{E}_{G(z,e)\sim\mathbb{P}_g}[min(0, -1 - D(C(G(z,e)), e))] \\
&- (1/2)\mathbb{E}_{x\sim\mathbb{P}_{mis}}[min(0, -1 - D(x, e))] \\
&+ k\mathbb{E}_{x\sim\mathbb{P}_r}[(||\nabla_x D(x, e)|| + ||\nabla_e D(x, e)||)^p]
\end{aligned}
$$

$$
L_G = -\mathbb{E}_{G(z,e)\sim\mathbb{P}_g}[D(C(G(z,e)), e)] - \lambda\mathbb{E}_{G(z,e)\sim\mathbb{P}_g}[S(G(z,e), e)] \tag{3}
$$

where $z$ is the noise vector sampled from $\mathcal{N}(0, 1)$ , $e$ is the sentence vector. k and p are two hyperparameters of gradient penalty; $\lambda$ is the coefficients of the text-image similarity. $\mathbb{P}_g, \mathbb{P}_r, \mathbb{P}_{mis}$ denote the synthetic data distribution, real data distribution, and mismatching data distribution, respectively

### 3.2.2 HARD MINING MATCHING-AWARE LOSS

Ignoring gradient penalty $\nabla$, the basic discriminator loss can be divided image-text matching-aware loss during synthetic data distribution $L_{fake}$, real data distribution $L_{real}$, and mismatching data distribution $L_{mis}$. For one training batchsize $N$, $L_D$ equals:

$$
\begin{aligned}
\sum_i^N L_{D_i} &= \sum_i^N L_{real}(x_i, e_i) + (1/2)\sum_i^N L_{fake}(\hat{x}_i, e_i) + (1/2)\sum_i^N L_{mis}(x_i, \hat{e}_i) + \nabla \\
&= (1/2)\sum_i^N (L_{real}(x_i, e_i) + L_{fake}(\hat{x}_i, e_i)) + (1/2)\sum_i^N (L_{real}(x_i, e_i) + L_{mis}(x_i, \hat{e}_i)) + \nabla \\
&= (1/2)\sum_i^N [\alpha - D(x_i, e_i) + D(\hat{x}_i, e_i)]_+ + (1/2)\sum_i^N [\alpha - D(x_i, e_i) + D(x_i, \hat{e}_i)]_+ + \nabla
\end{aligned}
$$

$$\tag{4}$$

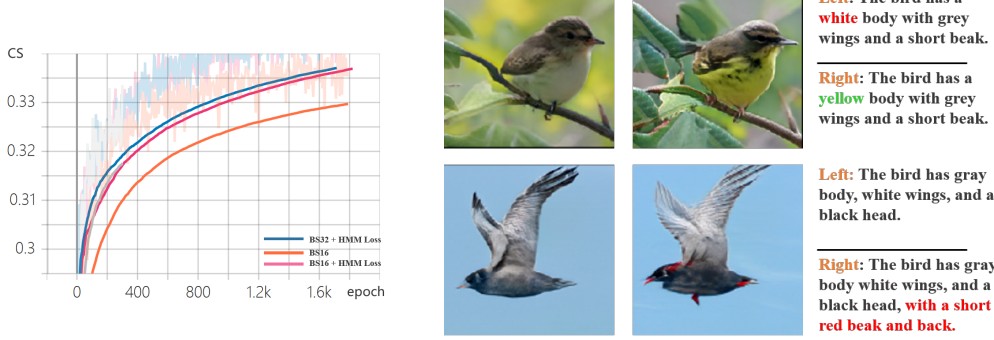

Figure 5: The Hard Mining Match-Aware loss in COCO training with different batch size.

Figure 6: Qualitative experiments on the CUB dataset, results of synthesized images with text descriptions changing.

For mismatching data common loss, referring to cross-modal retrieval (Faghri et al., 2017), We present a new change to loss functions for learning visual-semantic embeddings:

$$\hat{L_D} = (1/2) \sum_i^N [\alpha - D(x_i, e_i) + D(\hat{x}_i, e_i)]_+ + (1/2) \max_i [\alpha - D(x_i, e_i) + D(x_i, \hat{e}_i)]_+ + \nabla$$

$$(5)$$

Instead of finding hardest negatives in entire training set. We find them within each mini-batch, which can improve computational efficiency and reduce the dependence of generation quality on batchsize and epoch. But this modification is not suitable for synthetic data loss, because generating networks is an adversarial process. Hard negative mining of synthesized data loss will prematurely improve the discriminator's ability, resulting in the generator being unable to learn.

## 4  EXPERIMENTS

In this section, we introduce the datasets, and evaluation metrics, training setting, then evaluate our proposed HSPC-GAN method quantitatively and qualitatively to provide comparisons to the state-of-the-art.

### 4.1  EXPERIMENTS SETTING

**Datasets** Our experiments are conduct on two popular public datasets: CUB bird (Wah et al., 2011) and COCO (Lin et al., 2014). For the CUB bird dataset, there are 11,788 images belonging to 200 bird species, with each image corresponding to ten language descriptions. The cub provide rich attribute descriptions include shapes, colors, and postures et al., which is always employed to evaluate the performance of fine-grained content synthesis. COCO dataset contains 80000 images for training and 40000 images for testing, with each image provided to 5 image captions. The COCO image usually contains multiple objects under different scenes, which is employed to evaluate the performance of complex image synthesis. We use NLP tools to perform part of speech analysis on two datasets, extracting entities, attributes, and relationships described in the text to form new additional datasets. In the experiment, we replaced some descption texts for qualitative analysis, hoping that the model could focus on different information and generate images with different representations.

**Training and evaluation details** Follow the previous text-to-image works, we adopt the Fréchet Inception Distance (FID) (Heusel et al., 2017) and CLIP score (Wu et al., 2022) to evaluate the image fidelity and measure text-image semantic consistency. We also compared the convergence speed and training time of the model to prove that our loss function improvement is helpful for training. We choose the pretrained ViT-B/32 (Radford et al., ) as the CLIP model frozen with parameters. Based

on statistical experience in the dataset, the number of local entity prompts is limited to 6. Other parameter settings follow the basic model to ensure fairness. We employ the Adam optimizer with $\beta_1 = 0.0$ and $\beta_2 = 0.9$ to train our model. The learning rate is set to 0.0001 for the generator and 0.0004 for the discriminator. All models are trained on 8×3090 GPUs or 2xA800 GPUs.

## 4.2 QUANTITATIVE EVALUATION

Table 1: The results of FID and CLIP score (CS) compared with the state-of-the-art methods on the test set of CUB and COCO.

| Methods | Type | CUB | | COCO | |
|---|---|---|---|---|---|
| | | FID↓ | CS↑ | FID↓ | CS↑ |
| AttnGAN (Xu et al. 2018) | GAN | 23.98 | - | 35.49 | - |
| DM-GAN (Zhu et al. 2019) | GAN | 16.09 | - | 32.64 | - |
| XMC-GAN (Zhang et al. 2021) | GAN | - | - | 9.30 | - |
| DAE-GAN (Ruan et al. 2021) | GAN | 15.19 | - | 28.12 | - |
| DF-GAN (Tao et al. 2022) | GAN | 14.81 | 0.2920 | 19.32 | 0.2972 |
| LAFITE (Zhou et al. 2022) | GAN | 14.58 | 0.3125 | 16.09 | 0.3335 |
| VQ-Diffusion (Gu et al. 2022) | DM | 10.32 | - | 19.24 | - |
| LDM (Rombach et al. 2022) | DM | - | - | 12.63 | - |
| GigaGAN (Kang et al. 2023) | GAN | - | - | 9.18 | 0.307 |
| GALIP (Tao et al. 2023) | GAN | **10.08** | 0.3164 | 5.85 | 0.3338 |
| Baseline(GALIP*) | GAN | 11.2 | 0.3160 | 5.75 | 0.3385 |
| Ours | GAN | 10.11 | **0.3306** | **4.97** | **0.3507** |

To evaluate the performance of our HSPC-GAN, we follow the standard evaluation process and compare our method with several state-of-the-art methods, which have achieved impressive results in text-to-image synthesis. Table 1. shows the quantitative analysis results of our method compared to the current latest methods on the CUB dataset and COCO dataset. Compared with other leading models, our method has a significant improvement on both CUB and COCO datasets. Especially, compared to the latest methods, our HSPC-GAN improves the CLIP score (CS) from 0.3164 to 0.3306 on the CUB dataset and the CLIP score of COCO from 0.3338 to 0.3507. This indicates that our method can significantly improve the semantic consistency between text and image, and proves the effectiveness of the method based on hierarchical semantic prompts guided CLIP to align structured semantic concepts. With the recently proposed methods, GigaGAN and GALIP, which use GAN with the pre-training CLIP, our also has a significant lead in the COCO dataset. Our method decreases the FID metric from 5.85 to 4.97. Compared with diffusion model, VQ-Diffusion and LDM, we also have a better FID in the COCO dataset, which indicate that our synthesized image is closer to the semantics of the real image. To the FID of the CUB dataset, our baseline model is trained follow the parameter settings provided by the author. When the CLIP score is close, the FID do not reach the paper value, only 11.2. According to our method, the FID is reduced to 10.11 from 11.2, proving that our method is equally effective on the CUB dataset.

The influence of Hard Mining Matching-Aware Loss (HMMA Loss) is shown in Table 2. For the same training batch size, the loss can accelerate the convergence speed of the model and achieve better results. As Fig 5. shown, the mini training batch size adopt hard mining matching-aware loss can approach the learning effect in a large batch size than before. Additionally, experimental results show that when the hard mining matching-aware loss is applied to the synthesized sample, the generator will not be able to learn properly.

## 4.3 QUALITATIVE EVALUATION

Fig 1 shows the generated images of our method, which are compared with the traditional GAN network method SSA-GAN (Liao et al., 2022), other methods combining CLIP and GAN network GALIP, diffusion model SDM, and real samples. From this figure, we can find that CLIP can greatly improve the generation quality of traditional GAN networks, and our method focus on more fine-

Table 2: The impact of hard mining matching-aware loss during the model training on the CUB dataset. Epoch indicates the number of rounds for model to reach convergence, Max($\cdot$) indicates sample pair type of hard mining

| Loss Type | Batch Size | FID$\downarrow$ | CS$\uparrow$ | Epoch |
|---|---|---|---|---|
| base | 32 | 11.2 | 0.3160 | 1360 |
| base | 64 | 10.87 | 0.3171 | 1400 |
| Max(fake) | 32 | $+\infty$ | - | - |
| Max(fake)+Max(mis) | 32 | $+\infty$ | - | - |
| Max(mis) | 16 | 10.83 | 0.3185 | 1340 |
| Max(mis) | 32 | 10.87 | 0.3184 | 820 |
| Max(mis) | 64 | 10.7 | 0.3192 | 1100 |

grained semantic information in the context, such as the color of the bird's breast, the attribute of party "cocktail" and so on. These attribute information are always ignored by other methods. Fig.6 shows the generated results of our method when semantic changes occur. As the attributes described in the text change or increase, the visual information in the generated image will also change accordingly and other information remain unchanged. For example, only "white body" turns to "yello body", which indicates that hierarchical semantic prompts do indeed focus on changes in attribute information and affect the generated content based on the changes.

## 4.4 ABLATION STUDY

Table 3: The performance of different components of our model on the test CUB datasets

| Architecture | FID $\downarrow$ | CS $\uparrow$ |
|---|---|---|
| Baseline (with simple global prompt) | 11.2 | 0.3160 |
| +Local Prompts (6) | 10.83 | 0.3237 |
| +Local Prompts (4) | 10.97 | 0.3209 |
| +Hierarchical Semantic Prompts (6) | 10.51 | 0.3273 |
| +Hard Mining Matching-Aware Loss | 10.87 | 0.3184 |
| +Hierarchical Semantic Prompts (6), +Prompt CLIP Adaptor | 10.42 | 0.3284 |
| +Hierarchical Semantic Prompts (6), +Prompt CLIP Adaptor, +Hard Mining Matching-Aware Loss | **10.11** | **0.3306** |

To verify the effectiveness of each component in the proposed HSPC-GAN, we conduct ablation studies on the test set of the CUB dataset. The components being evaluated in this subsection include different Prompts and Hierarchical Semantic Prompts (HSP). We also test the CLIP adaptor and hard mining matching-mware Loss.As shown in Table 3., replacing simple global prompts with hierarchical semantic prompts can introduce more contextual semantic information. Selective adjustment of CLIP improves the semantic consistency of synthesized images. Hard mining Matching-Aware Loss results in further performance improvement. In addition, maximum number of entity prompts has an impact on the experimental results, becuase fewer entity prompts will limit the extraction and alignment of relational semantics. And more relevant ablation experiment results will be presented in the appendix.

## 5 CONCLUSION

In this work, we propose HSPC-GAN for the text-to-image generation tasks. We present a construction method of hierarchical semantics prompts and introduce how to use hierarchical prompts to selectively adjust visual features in order by freezing CLIP parameters. Also, we propose a novel Hard Mining Matching-Aware Loss in mis-sample for image generation. It can further enhance the text-timage semantic consistency, accelerate training, and reduce the demand for computing resources. Extensive experiment results demonstrate that our method HSPC-GAN significantly outperforms current state-of-the-art models on the CUB dataset and COCO dataset.

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

## A  APPENDIX

We provide other additional sections Here.

### A.1  THE INFLUENCE OF HIERARCHICAL SEMANTIC PROMPT ORDER

Table 4: The influence of hierarchical Semantic Prompt order, G is global prompt, L is local prompt. The results on the CUB dataset

| Prompt Type | FID↓ | CS↑ |
|---|---|---|
| only G | 10.71 | 0.3174 |
| only L | 10.67 | 0.3182 |
| L, G | 10.55 | 0.3174 |
| G, L | 10.39 | 0.3200 |
| G ,L ,G | 10.11 | 0.3306 |

We discuss how Prompts use different combinations and orders to guide CLIP adaptors in adjusting the impact of image features on the generated results. The experiment is tested on the CUB dataset with detailed attribute descriptions.The results are shown in Table 4. Using only global or local

prompts both can cause semantic information loss, and attaching global prompts before and after local prompts can better ensure the semantical structure of visual feature information.

## A.2 COMPARISON OF SYNTHESIZED IMAGES WITH OTHER METHODS

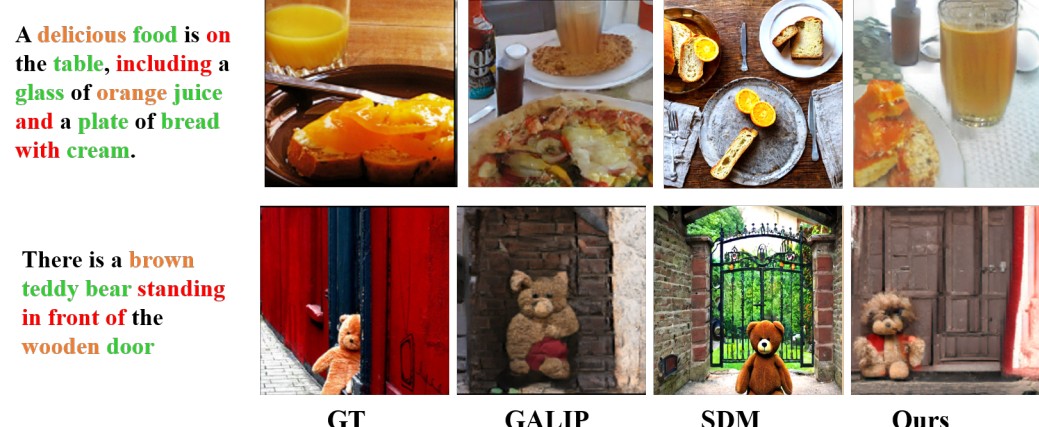

**A delicious food is on the table, including a glass of orange juice and a plate of bread with cream.**

**There is a brown teddy bear standing in front of the wooden door**

Figure 7: Comparisons of image generation results with different methods on the COCO dataset

The fig.7 shows more comparisons between our results and other motheds on the COCO dataset , include GALIP, Stable Diffusion Model. Compared to other methods, we focus more on the detailed semantic information in the text. For example, the entity "juice" and "door", the attribute "wooden".

