# OpenReview forum: "generative adversarial network with hierarchical semantic prompt constrainting clip for high-quality text-to-image synthesis"
_ICLR.cc/2024/Conference — ICLR 2024 Conference Withdrawn Submission_

### Official Review · Reviewer_hvaD · 2023-10-29

**Soundness:** 1 poor
**Presentation:** 1 poor
**Contribution:** 1 poor
**Rating:** 1
**Confidence:** 4

**Summary:**

The diffusion model achieves compelling results on text-to-image tasks, but its efficiency is far from satisfactory, particularly lagging behind GAN models. Prior attempts treat CLIP as a text encoder and naively combine it with GAN, leading to undesirable images that are semantically inconsistent with text prompts. This manuscript, HSPC-GAN, leverages a prompt generator and a prompt adaptor to construct structural-semantic prompts for guiding CLIP to adjust the visual features of GAN models. Some quantitative results validate the effectiveness of such a subtle design.

**Strengths:**

The design of using structural-semantic prompts for better and more consistent visual features is somewhat reasonable.

**Weaknesses:**

1. This paper is definitely finished in a rush. The whole manuscript is full of various grammar mistakes, spelling and punctuation errors, and unnatural expressions. For example, the first sentence of the abstract is weird. It should be “How to efficiently synthesize controllable, semantically-aligned and high-quality images based on text prompts is currently a significantly challenging task”. Moreover, there are a lot of spelling and punctuation errors in the sections of the introduction and experiment, making the reviewer so hard to understand.
2. The authors hardly provide qualitative comparisons between their method and baselines. Only a few examples in Fig 1,6&7 can demonstrate the superiority of their proposed method. Actually, the experiment section should present enough synthesized images.
3. The quantitative results listed in Table 1 show the proposed HSPC-GAN can achieve minor improvement over several baselines, in terms of FID and CLIP score. However, this manuscript always emphasizes semantic alignment, which obviously can’t be reflected with such two metrics.

**Questions:**

Why does the result of GALIP appear twice in Table 1?

---

> ### Author Response · Authors · 2023-11-22
>
> Firstly, we acknowledge and will work to improve the expression issues caused by the haste in writing.
>
> Regarding qualitative comparisons, although there are fewer images, each experiment includes qualitative comparisons. It is considered unreasonable by the reviewer to claim there are almost none; nonetheless, we will add more synthetic images to address this concern.
>
> As for the third issue, we find it somewhat confusing. Firstly, we are unsure if the reviewer has thoroughly read the entire paper, because the question “Why does the result of GALIP appear twice in Table 1?”  which has been explained in the paper. One is the literature result, and the other is the baseline we actually reproduced, and the results are different.
>
> Secondly, FID is used to measure the gap between synthetic images and the originals, while CLIPScore is an indicator in the image synthesis field for evaluating the consistency between images and text. The reviewer questions the ability of CLIPScore to measure semantic alignment between images and text, in other words, questioning the rationality of employing a universal evaluation metric for the relevant domain, but does not provide sufficient reasons. This raises doubts about the reviewer's expertise in this field.

---

### Official Review · Reviewer_v8PS · 2023-10-30

**Soundness:** 2 fair
**Presentation:** 1 poor
**Contribution:** 2 fair
**Rating:** 3
**Confidence:** 4

**Summary:**

The paper proposes a new GAN model for text-to-image synthesis, called HSPC-GAN. The model is based on GALIP [Tao et al, CVPR23]. and extends this baseline with several modules like text concept mining, hierarchical semantic prompt generation, hierarchical semantic prompt adaptor, hard mining matching-aware loss. The model is evaluated on the task of text-to-image synthesis using CUB and COCO datasets. HSPC-GAN outperforms the baseline GALIP, as well as other GAN and DM baselines, in FID and CLIP scores.

**Strengths:**

- The proposed model achieves visually good performance of text-to-image synthesis. This establishes a strong and relatively light-weight GAN baseline in the task which is dominated by heavy diffusion models with much longer sampling time.

- The improvement over the baseline GALIP is visible in both FID (image quality) and CLIP score (text-image alignment).

**Weaknesses:**

- $\underline{\text{Presentation}}$. I think the major weakness of this submission is presentation.
  - The whole paper dives deep into technical implementation of the proposed method, but lacks high-level explanations and discussions of contributions. What are the key issues with prior GAN-based text-to-image methods, and why the proposed method is the most efficient way to address them? What could be the alternatives for the proposed extensions? What are the limitations of using all the pre-trained modules (e.g., Text Concept Mining or Scene Graph Generator)? These questions are not answered. It is thus not clear what high-level lessons can a reader extract.
  - The paper is full of typos, and non-standard terminology. Examples of typos include: "pre-trainde", "demonstrat", "not not excellent", "adptopted", "As Fig 5. shown", "matching-mware Loss", "CLIP ADPTOR", "experiments are conduct". I think the paper needs careful revision regarding writing.
  - Why does "GAN-based image synthesis" section of the related work discuss GAN inversion? (This is not a paper about GAN inversion). On the other hand, why is there no discussion of other text-to-image GAN and DMs?

- $\underline{\text{Novelty}}$. It is not clear what kind of technical novelty does the paper present. In particular, the method section starts from an existing baseline GALIP. Next, the method combines modules that existed in some forms in other fields, or slightly modifying the D loss, without analysing the challenges that arise during these modifications. It is not clear what is the technical novelty in the paper, what is the technical challenge that was addressed, and what is the technical lesson for the readers.

- $\underline{\text{Unsupported claim}}$. "This loss can be generalized to apply most generation methods involving semantic
matching. We will do more work in the future to prove this viewpoint." - This is a rather strong claim that has no confirmation in the experiments. The form of the loss is binded to the GAN-form objective and it is in fact not clear if it would generalize to any other objectives.

- $\underline{\text{Results}}$. Given the nature of this line of research, the visual results in this paper are limited (4 images in the main paper and 2 in the supplementary).

- $\underline{\text{Efficiency}}$. At a global level, the paper introduced several new modules on top of GALIP model. The new model achieves somewhat better FID and CLIP scores but works noticeably slower (e.g., 0.025s vs 0.017s in Fig. 1). The efficiency aspect and possible trade-offs are not discussed in the paper.

**Questions:**

Please answer the comments from the Weaknesses section.

---

> ### Author Response · Authors · 2023-11-22
>
> Firstly, thank you very much for your feedback. Regarding the expression issues, due to the short writing time during the submission of this article, there were several typos, and some discussions became unclear after being cut due to length constraints. If the paper is accepted, we will carefully revise and rectify these expression issues in the final version.
>
> For your other questions, we provide detailed responses as follows:
>
> 1.The key issues in previous GAN-based text-to-image methods are twofold. Firstly, they are constrained by modality gaps and dataset limitations, making it challenging for GANs to find a good balance point in the latent space for effective feature mapping between text and images. This can be addressed through the use of pre-trained CLIP. Secondly, there is a challenge in semantic constraints on synthesized images – ensuring the consistency between image and text content, which is the focus of our approach. We control this using hierarchical semantic prompts. The choice of hierarchical prompts as a solution stems from their effective use in controlling prompts in text generation, and we aim for a similar effect in image generation.
>
> 2.Regarding the GAN inversion part, this was an oversight on our part. In the initial version, we discussed the correlation between constructing hierarchical semantic prompts and GAN inversion in the modal latent space, and how it influences and interacts with the generator. We failed to adjust this section after its removal in the submitted version, and we will rectify that.
>
> 3.Concerning the novelty of our work, we indicate in the paper that the innovation lies in:
>
> （1）How to construct appropriate semantic prompts from text to guide the synthesis of target and attribute information in images. The current construction of prompts in this field is rough, if not overlooked.
> （2）How to use different hierarchical semantic prompts to constrain the adjustment of target synthesis in the generator. This involves determining which prompts to use, which to ignore, and how to adjust the generator's output.
> （3）The introduction of hard-negative sample loss in the GAN network is a key aspect.
> Regarding the viewpoint that challenges during the analysis of loss modifications were not addressed, we disagree. Whether it's the construction of hierarchical semantic prompts, feature adjustments based on prompts, or the introduction of hard-negative samples in the discriminator, we have analyzed the purpose of these approaches, the problems they aim to solve, the difficulties in their resolution, and the advantages of adopting such methods.
> Certainly, if you feel that our explanation is not clear enough, we can also supplement more content in the formal version's appendix to provide a more detailed explanation of our methods and analysis. Thank you for your feedback; we will strive to ensure that readers have a comprehensive understanding of our work.
>
> 4.Regarding the “unsupported claim”，we think this statement is not a strong claim but rather an outlook and consideration for future work.  the new discriminative loss applied in the hinge loss form, we drew inspiration from cross-modal retrieval work and redesigned a new discriminative loss for image matching between synthetic images, real images, and mismatched images in the GAN network. The experiments in the paper demonstrate the effectiveness of the final loss structure. Since the use of hinge loss in image matching is not exclusive to GANs but also includes VAE and some diffusion models in image synthesis, we point out the potential for its generalization to other synthetic methods with a similar structure.
>
> 5.Regarding image issues, we will supplement more image examples in the final version.
>
> 6.Regarding the efficiency trade-off, this issue is commonly present in large-scale image synthesis tasks, therefore we aim to pursue higher synthesis quality within the same efficiency scale (0.001s, 0.01s, 0.1s).

---

### Official Review · Reviewer_rLWE · 2023-10-31

**Soundness:** 2 fair
**Presentation:** 2 fair
**Contribution:** 2 fair
**Rating:** 5
**Confidence:** 3

**Summary:**

This work introduces HSPC-GAN,  a novel text-to-image synthesis framework that combines GANs with CLIP models to produce high-quality images that are semantically consistent with textual descriptions. It introduces structured semantic prompts for more accurate visual feature adjustments and a new loss function for improved training efficiency. HSPC-GAN achieves state-of-the-art results, synthesizing images faster and with greater semantic relevance compared to existing methods.

**Strengths:**

It leverages the semantic alignment strengths of CLIP to enhance the quality of images generated by GANs.
By analyzing text to identify entities, attributes, and relationships, the model can generate images that are not only visually appealing but also semantically consistent with the input text.

**Weaknesses:**

The main concern is the effectiveness of the proposed designs.
As shown in the ablation study of Table 3, the results are not convincing for me.
For example, comparing '+Hierarchical Semantic Prompts (6)' and '+Hierarchical Semantic Prompts (6), +Prompt CLIP Adaptor', it seems Prompt CLIP Adaptor can only improve the FID by 0.09, CS by 0.0011, which is very marginal and could be neglected considering the seed randomness.

Besides, it seems '+Hard Mining Matching-Aware Loss' can improve the baseline by the 0.33 FID and 0.0024 CS. But when applying it to a stronger model '+Hierarchical Semantic Prompts (6), +Prompt CLIP Adaptor', the FID and CS can still be improved 0.31 FID and 0.0022. It is not convincing for me as the improvement should be clearly narrowed with a stronger base model.

**Questions:**

See above.

---

> ### Author Response · Authors · 2023-11-22
>
> Firstly, thank you for your feedback. Indeed, there are some shortcomings in the design of our ablation experiment table. If the paper passes the review, we will add more supplementary explanations in the final version.
>
> Regarding the two questions you raised, firstly, the issue of the seemingly limited improvement in the CLIP adapter in the ablation experiment. In the table, the role of the CLIP adapter is to filter and control semantic prompts, serving the adjustment of image features by hierarchical semantic prompts, as shown in Figure 4b. The use of gating is employed to adjust and filter local and redundant prompts. In the case of using only the CLIP adapter, the FID value is 10.66 (apologies for not including it in the table, we will add it later). The hierarchical semantic prompts (6) + prompt CLIP adapter together achieve the adjustment of the CLIP adapter and contribute to a 0.78 improvement in FID. The use of these two modules does not simply result in a 1+1>2 effect; instead, they mutually influence each other. Higher-quality semantic prompts can weaken the effect of the adapter, hence the marginal improvement.
>
> Regarding the second question, the hard-mining matching-aware loss is primarily employed to address the handling of challenging negative samples within the discriminator network in a batch, aiming to expedite model training. Changes in both hierarchical semantic prompts and prompt adapters are designed for the generator to enhance the semantic consistency between text and images. They do not involve modifications to the discriminator, and there is no inherent correlation between the two. Therefore, the marginal effects are not pronounced (FID improvement from 0.33 to 0.31, and CS from 0.0024 to 0.0022). According to relevant literature on cross-modal retrieval, such improvements are more susceptible to the dataset and discriminator loss. If you have further questions on this matter, after the paper is accepted, we plan to open-source the relevant code.
>
> Second，

---

### Official Review · Reviewer_rmDR · 2023-11-08

**Soundness:** 2 fair
**Presentation:** 3 good
**Contribution:** 2 fair
**Rating:** 3
**Confidence:** 5

**Summary:**

This paper proposes HSPC-GAN, a method of constructing structural semantic prompts and using them to hierarchically guide CLIP to adjust visual features for generation of high-quality images with controllable semantic consistency. HSPC-GAN extracts semantic concepts through part of speech analysis, constructs a prompt generator and a prompt adaptor to generate learnable hierarchical semantic prompts, and using these prompts to selectively guide CLIP adapters to adjust image features to improve semantic consistency between synthesized images and conditional texts.

**Strengths:**

This article is well-written and easy to understand.

**Weaknesses:**

The experiments are not sufficient to prove the effectiveness of the method in the paper. For example,
1. Compared with the GALIP paper, there is a lack of comparison results of Model Param and Data Size.
2. Compared with the GigaGAN paper, there is a lack of comparison results of high-resolution generation.

**Questions:**

See Weaknesses

---

> ### Author Response · Authors · 2023-11-22
>
> Q1:Compared with the GALIP paper, there is a lack of comparison results of Model Param and Data Size.
>
> A1:My model's parameters are 0.3+0.08B, while GALIP's parameters are 0.24+0.08B. The additional parameters are used for generating structural semantic prompts and hierarchical adjustment in the generator. The parameter count is much smaller than that of diffusion generative models.
> We have provided details about the data size in the training section. We adopted GALIP's training settings (includeing the same data size)and trained only on the standard MSCOCO dataset (80000 images).and CUB_BIRD(11,788 Images)
>
> Q2:Compared with the GigaGAN paper, there is a lack of comparison results of high-resolution generation.
>
> A2:Firstly, the primary focus of this paper is to investigate the semantic consistency between text content and synthesized image content; high-resolution image synthesis is not the focal point of this study. We utilized the basic framework of GALIP, which itself is not suitable for synthesizing ultra-high-resolution images. This is in contrast to the foundational model of GIGAGAN, which adopts the basic framework of StyleGAN and incorporates a design for a super-resolution module to synthesize higher-resolution images post the standard model. Therefore, we did not provide a comparison of super-resolved images in the paper because we believe it is not directly relevant to our work. If you deem it necessary and are willing to modify the rating, we would be willing to include comparisons of higher-resolution images in the appendix after the paper is accepted.

---

### Meta-Review · Area_Chair_q9mY · 2023-12-05

**Metareview:**

All reviewers recommend rejecting the paper. The reviewers raise concerns about insufficient experiments, limited result quality, and poor presentation. The authors are encouraged to resolve these issues and submit to another venue.

**Justification For Why Not Higher Score:**

Reviewers raised several concerns summarized in the meta-review, which the authors are encouraged to address.

**Justification For Why Not Lower Score:**

N/A

---

### Decision · Program_Chairs · 2024-01-16

Reject